# Polarization Z-Scan Studies Revealing Plasmon Coupling Enhancement Due to Dimer Formation of Gold Nanoparticles in Nematic Liquid Crystals

**DOI:** 10.3390/mi14122206

**Published:** 2023-12-05

**Authors:** Shengwei Wang, Edward J. Lipchus, Mohamed Amine Gharbi, Chandra S. Yelleswarapu

**Affiliations:** Department of Physics, University of Massachusetts Boston, 100 Morrissey Blvd, Boston, MA 02125, USA; shengwei.wang001@umb.edu (S.W.); edward.lipchus001@umb.edu (E.J.L.)

**Keywords:** plasmon coupling, nanoparticle dimers, gold nanoparticles, nonlinear absorption, nematic liquid crystals, polarization z-scan, plasmon enhancement, third-order nonlinear absorption

## Abstract

We investigate the plasmon coupling of gold nanoparticle (AuNP) dimers dispersed in a nematic liquid crystal matrix using the polarization z-scan technique. Our experimental setup includes the precise control of incident light polarization through polarization angles of 0°, 45°, and 90°. Two distinct cell orientations are examined: parallel and twisted nematic cells. In parallel-oriented cells, where liquid crystal molecules and AuNPs align with the rubbing direction, we observe a remarkable 2–3-fold increase in the nonlinear absorption coefficient when the polarization of the incident light is parallel to the rubbing direction. Additionally, a linear decrease in the third-order nonlinear absorption coefficient is noted as the polarization angle varies from 0° to 90°. In the case of twisted nematic cells, the NPs do not have any preferred orientation, and the enhancement remains consistent across all polarization angles. These findings conclusively establish that the observed enhancement in the nonlinear absorption coefficient is a direct consequence of plasmon coupling, shedding light on the intricate interplay between plasmonic nanostructures and liquid crystal matrices.

## 1. Introduction

Noble metal nanoparticles (NMNPs) exhibit a fascinating phenomenon when interacting with electromagnetic radiation, resulting in localized surface plasmon resonances (LSPR) [1,2,3]. When these NMNPs are dispersed within a dielectric medium, LSPR can enhance local electromagnetic fields and modify the dielectric properties of the surrounding medium [4,5]. The collective oscillations of conduction electrons align with the amplitude of the electromagnetic radiation [6]. When two NMNPs are placed in close proximity, they form a dimer. If there is an interaction between the collective oscillations, plasmonic oscillations can couple through near-field interactions, further enhancing the electromagnetic field and modifying the optical properties of the dielectric medium. This phenomenon has found applications in various research fields, including materials science, nanotechnology, and optics [7,8,9,10]. The properties of dimers can be tuned by controlling parameters such as nanoparticle size, shape, composition, and interparticle spacing [11,12,13,14].

A variety of techniques are employed in the synthesis of nanoparticle dimers, including chemical synthesis [15], ligand-mediated self-assembly [16], template assembly, and the dispersion of colloidal particles in liquid crystals [13,17,18]. Among these approaches, utilizing liquid crystals for arrangement offers a simpler and more adaptable alignment method [19,20,21,22,23]. AuNPs are introduced into planar nematic liquid crystals (NLCs) with an aligned director, resulting in an observed increase in photoluminescence (PL) within a specific range of AuNP concentrations. Moreover, the PL decreases when rotating the director of the samples from parallel to perpendicular under polarized light [24]. In another approach, colloidal microspheres are used to induce topological defects in nematic liquid crystals, which provide a scaffold for plasmonic nanoparticles [25]. Recently, the ordering of anisotropic nanoparticles within liquid crystalline colloidal hosts has been harnessed to achieve controlled three-dimensional (3D) coalignment of gold nanorods and cellulose nanocrystals [26].

The incorporation of various nanoscale materials, including magnetic and ferroelectric nanoparticles, carbon nanotubes, and nanosolids, into twisted nematic liquid crystals (TNLCs) has also been a subject of investigation. The results revealed that these materials can effectively manipulate the electrical and magnetic properties of TNLC, thanks to the efficient templating provided by the TNLC structure [27,28,29,30,31]. An efficient method for inducing twisting in NLCs involves orienting the rubbing directions of anchoring materials on two substrates perpendicular to each other [32]. Recently, we demonstrated enhanced third-order nonlinear absorption when AuNPs are dispersed in planar-oriented nematic liquid crystals (4′-Pentyl-4-biphenylcarbonitrile-5CB) [33].

In addition to the formation of nanoparticle dimers, it is crucial to elucidate their arrangement and plasmon coupling for comprehensive understanding and optimization. Various techniques, including surface-enhanced Raman scattering, UV–VIS and scattering spectroscopy, as well as fluorescence resonance energy transfer, among others, have been utilized to investigate the organization of nanoparticle dimers and plasmon coupling. For instance, Lamprecht et al. discovered that the optical properties of two-dimensional arrays of gold nanoparticles rely on interparticle distance and the polarization direction of incident light. They also noted a redshift of the plasmon resonance wavelength and an increase in plasmon damping in the transition region between short-range and long-range coupling regimes [34]. Danckwerts’s team found that nonlinear-optical four-wave mixing is significantly enhanced as the gap size between two NPs decreases, particularly when reaching the touching range [12].

Polarization z-scan allows for a comprehensive analysis of the material’s anisotropic properties, revealing how nonlinear optical behavior changes with different polarizations [35,36,37,38,39,40,41]. In this report, we used the polarization z-scan technique to probe the interparticle plasmon coupling within AuNP dimers as well as establish their presence and structural characteristics. This was performed by rotating the polarization of light to excite the AuNPs at different angles and measuring the nonlinear optical response of the liquid crystal medium. Our findings reveal that when the plasmon oscillation aligns with the plane of polarization, a notable enhancement in third-order nonlinear absorption occurs because of plasmon coupling.

## 2. Materials and Methods

### 2.1. Sample Preparation

AuNPs were initially suspended in water. To transfer them into 5CB nematic liquid crystals (4′-Pentyl-4-biphenylcarbonitrile), we first evaporated the water, and then the NPs were redissolved in ethanol. Once the desired quantity was achieved, they were added to 5CB. Subsequently, the mixture was heated and stirred until the ethanol completely evaporated. The resulting AuNP/5CB solution was then injected into prefabricated planar-oriented cells. The preparation of these cells involved several stages. Initially, two glass plates (25 × 20 × 1.0 mm, Fisher brand) were cleaned with water and acid, followed by coating them with a monolayer of 3 wt% polyvinyl alcohol (PVA) solution (90% water + 10% ethanol + 3% PVA). The plates were then crosslinked by placing them in an oven at 110 °C for 1 h to ensure planar anchoring via the PVA film. To achieve planar-oriented anchoring of the liquid crystal molecules, the glass plates were uniformly rubbed with cloth to impart a preferred direction onto the PVA film. A 20 µm Mylar spacer (Grafix wrap) was inserted between the glass plates to set the cell thickness, and the cell was sealed with optical glue. For twisted nematic cells, the rubbed glass plates were placed perpendicular to each other. A 0.05 wt% AuNP (NN-Labs)/5CB mixture was introduced into the cells through capillarity while heating the mixture to the isotropic phase of 5CB to prevent elasticity and NP aggregation.

### 2.2. Polarization Z-Scan Setup

A schematic of the open aperture z-scan system is shown in Figure 1. The laser system comprises a frequency-doubled Nd:YAG laser (Continuum Minilite II) emitting at 532 nm with a pulse width of 3–5 ns and a beam waist of 55 μm. A 20 mm lens is used to focus the laser beam, and the beam intensity is kept constant for all z-scan measurements. The AuNP/5CB liquid crystal cell is mounted on a translation stage (Thorlabs NRT 150) and moved freely along the *z*-axis, passing through the laser beam’s focal point. The transmittance data are collected using an optical detector (Newport 818-SL), while the translation of the stage and data acquisition are meticulously controlled through a LabView routine. Additionally, the setup consists of a polarizer and a half-wave plate (HWP) to manipulate the polarization orientation of the laser beam. Despite the laser’s initial vertical polarization, a polarizer is introduced for ease of alignment. Notably, the angle between the input and output polarizations of the HWP is twice the angle between the input polarization (i.e., the polarizer) and the fast axis of the HWP. As a result, by setting the angle between the polarizer and the HWP’s fast axis to 45°, the HWP plate achieves a 90° rotation, converting vertically polarized light into horizontal polarization. Thus, to achieve polarization angles of 0°, 45°, and 90°, the angle between the polarizer and the fast axis of the HWP is meticulously adjusted to 0°, 22.5°, and 45°, respectively.

## 3. Results and Discussion

When liquid crystal molecules meet the rubbed PVA surface, their long axes align parallel to the rubbing direction. This alignment occurs due to the anisotropic nature of the liquid crystal molecules and the surface properties of the substrate. Two types of NLC samples are prepared: parallel and twisted cells, as shown in Figure 2. In the case of parallel-oriented cells, the rubbing directions on the two glass plates are parallel to each other, and, therefore, the alignment of the long axes of the liquid crystal molecules tends to align parallel to the rubbing direction on both surfaces. This alignment gives rise to an anisotropic structure, where the properties of the liquid crystals vary with direction. In the twisted nematic liquid crystal cell case, the rubbing directions are perpendicular to one another, such that the liquid crystal molecules are aligned at one substrate, and they twist gradually until they reach alignment at the opposite substrate. This twist creates a helical structure throughout the liquid crystal layer.

Figure 3 show cross-polarized optical microscopy images of parallel and twisted nematic liquid crystal cells, respectively. When the easy axis of the cell is parallel or perpendicular to one of the polarizers (Figure 3a, left), no light is observed referring to the perfect alignment of the liquid crystal cell. If we rotate the cell and align its easy axis at 45° with the polarizer and analyzer (Figure 3a, right), light will pass through the sample, confirming good alignment of the director. In the case of twisted cells (shown in Figure 3b), light can be observed regardless of the orientation of the cell. Since the NPs dimers orient themselves with the nematic director, these dimers should have the corresponding orientation in the oriented and twisted cells, as shown in the schemes of Figure 2.

Third-order nonlinear optical absorption studies were carried out using the open-aperture z-scan technique for pure 5CB, parallel, and twisted AuNP/5CB samples. For each sample, z-scans were obtained by setting the laser beam polarization at 0°, 45°, and 90°, as shown in Figure 2. The third-order nonlinear absorption coefficient (β) is obtained by curve fitting the normalized transmittance data from the z-scan using Equation (1) [42], where the linear absorption coefficient α is obtained from the absorbance spectrum (see Figure A1 and Figure A2 in Appendix A), using the Beer–Lambert law, and L is the sample thickness (20 μm). I_0_ is the on-axis light intensity (which is fixed to all z-scan processes), and z_0_ is the Rayleigh range, where ω_0_ and λ are the beam waist and wavelength of the laser source, respectively.
(1)$$T\left(z\right)=\sum_{m=0}^{\infty }\frac{{\left[-\beta L_{eff}\frac{I_{0}}{\left(1+\frac{z^{2}}{z_{0}^{2}}\right)}\right]}^{m}}{{\left(m+1\right)}^{\frac{3}{2}}},\;L_{eff}=\frac{1-e^{-\alpha L}}{\alpha }$$

The absolute and relative values of *β* for all the samples are shown in Figure 4. The *β* values for the pure 5CB samples are much lower and are independent of the polarization. However, the *β* values for the AuNP/5CB samples are 2–3 times higher and depend on the polarization of the incident light and the type of liquid crystal cell. For the parallel-oriented AuNP/5CB cell, the *β* values decrease as the polarization of the incident light is changed from 0° to 90° with respect to the rubbing surface, as shown in Figure 4a. The effect is the same but reversed when the cell is rotated by 90° (see Figure A3 and Figure A4 in Appendix A). In both cases, *β* is optimum when the polarization of the incident light is parallel to the rubbing direction.

When AuNPs are dispersed in 5CB, elastic forces cause the dispersed AuNPs to align parallel to the 5CB director to minimize elastic distortions brought about by the presence of particles. Thus, AuNPs follow the long axes of the liquid crystal molecules. In parallel liquid crystal cells, AuNPs are aligned parallel to the rubbing direction on both surfaces. Thus, AuNP dimers are formed along the rubbing direction, and when the electric field direction is parallel to the AuNPs dimer, as shown in Figure 5, the LSPR oscillations will overlap and enhance the electromagnetic field. When the polarized light is incident at other angles, the component along the dimer will be less and so the field will be weaker.

In the case of a TNLC cell, AuNPs are aligned vertically (let us say) at one substrate, and they twist gradually until they reach the horizontal alignment at the opposite substrate. This twist creates a helical structure throughout the liquid crystal layer, as depicted in Figure 2. This alignment has an equal effect for all three 0°, 45°, and 90° polarization angles of incident light, as shown in Figure 4a. The β values are enhanced by a factor of two for twisted AuNPs/5CB compared to 5CB (control group). Figure 4b shows the trend in relative β values normalized to the β values obtained when the incident light is polarized at 45°. The linear decrease in relative β values for the vertical sample shows a distinct lessening of plasmon coupling and indicates a clear orientation of the dimers along the rubbing direction. Conversely, the uniform trend from the twisted sample indicates no overall orientation of the dimer as it displays equal enhancement in all polarizations. When the liquid cell orientation is rotated by 90°, the relative β value trend increases linearly, as shown in Figure A4 in Appendix A.

Thus, when the polarization of the incident light parallel to the liquid crystal cells, the interaction between the AuNPs and incident light will be influenced by the uniform alignment of the liquid crystal molecules, potentially enhancing the plasmon resonance of the NPs due to the ordered alignment of the liquid crystal molecules. In a TNLC cell, where the liquid crystal molecules undergo a gradual twist across the thickness of the cell, the interaction between the AuNPs and the incident light is averaged by the changing molecular alignment.

This observation can be explained by considering the formation of AuNP dimers and their interaction with the incident light. We also modeled the dimer coupling effect across orientations using COMSOL finite element analysis using the Drude-Lorentz dispersion model:(2)$$\varepsilon_{r}=\varepsilon_{\infty }+\overset{M}{\underset{j=1}{\sum }}\frac{f_{j}\omega_{p}^{2}}{\omega_{0_{j}}^{2}-\omega^{2}+i\Gamma_{j}\omega },\sigma =0$$
for the Drude dielectric constant $\varepsilon_{r}$ used for the refractive index for the gold nanoparticles, the high-frequency relative permittivity $\varepsilon_{\infty }=1$, the oscillator strength *f* = 1, the plasma frequency of gold $\omega_{p}$ = 1.37 × 10^16^ rad/s, the resonant frequency of gold at this size $\omega_{0}$ = 3.80 × 10^15^ rad/s, and the bulk damping constant for gold Γ = 2.08 × 10^14^ rad/s. The conductivity was set to zero, and the relative permeability was set to one.

In this simulation, two 10 nm radius gold nanoparticles were suspended in aqueous solution and excited by a 520 nm wavelength plane wave incident along the *z*-axis and polarized along the *x*-axis. The amplitude of the electric field was 1 V/m, and instead of rotating the polarization of the light, the relative positions of the nanoparticles were adjusted to allow for easier comparisons. Field enhancement of the localized electric field was observed for a series of different separation distances, but presented here are the results for a separation of 2 nm between the spheres. Figure 5 shows the norm of the electric field in local space across a *xy*-plane cut for the three relative orientations (0°, 45°, and 90° to the incident light). Since the incident field strength was set to 1 V/m, these values represent the field enhancement. This is also visualized as a height map of electric field intensity. Therefore, from Figure 4 and Figure 5, it can be concluded that when oriented parallel to the incident light, the AuNPs show strong coupling, which lessens at 45° and does not exist at the 90° orientation.

The plot in Figure 6 shows the field enhancements and their normalization against 45°, and then compared against the normalized β values from the experiment. At 0°, the dimer exhibits roughly three times the field enhancement at 90°. With no coupling, while aligned perpendicular to the incident light, the AuNPs act as individual particles, and as such, this change in field enhancement is in line with expected values [8]. When normalized to 45° to compare against the β values from the experiment, we see that they are almost identical.

To ensure that the AuNP/5CB cells have no spatial variance, we performed the z-scan at various locations on the sample. Figure 7 shows the plot between β values for single and multiple locations as the polarization of the laser beam is varied. For the single location, we performed multiple measurements (run many times and obtained group β values) to avoid a single accidental result. The decreasing trend of the β values when the polarized light is rotated from parallel to perpendicular to the dimers is consistent.

## 4. Conclusions

In conclusion, we conducted a study on the plasmon oscillations of AuNP dimers dispersed in 5CB nematic liquid crystals and examined their coupling using polarization z-scans. A polarizer and a half-wave plate (HWP) were used to control the laser beam’s polarization orientation, and the HWP was adjusted to achieve polarization angles of 0°, 45°, and 90°. We prepared two types of planar-oriented samples: parallel and twisted nematic liquid crystal cells. In parallel-oriented cells, the long axes of the liquid crystal molecules and AuNPs align in parallel with the rubbing direction on both surfaces. In the case of twisted nematic liquid crystal cells, both the long axes of the liquid crystal molecules and AuNPs gradually twist by 90° from one end to the other within the cell. The results demonstrate that in the case of parallel-oriented cells, when the incident light polarization aligns with the rubbing direction, the nonlinear absorption coefficient is approximately 2–3 times greater than that of 5CB alone. Additionally, it is observed that the value of the nonlinear absorption coefficient decreases linearly as the polarization angle changes from 0° to 90°. Conversely, for twisted nematic cells, the enhancement remains consistent across all polarization angles. These findings confirm that the enhancement of the nonlinear absorption coefficient is indeed attributed to plasmon coupling. They also show that the polarization z-scan technique can be a powerful tool to verify the alignment of nanomaterials in anisotropic materials.

## Figures and Tables

**Figure 1 micromachines-14-02206-f001:**
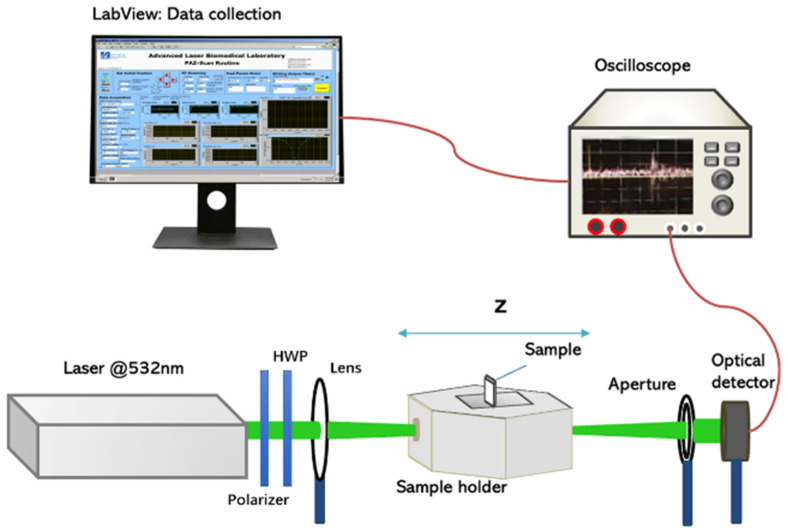
Schematic of the open aperture polarization z-scan.

**Figure 2 micromachines-14-02206-f002:**
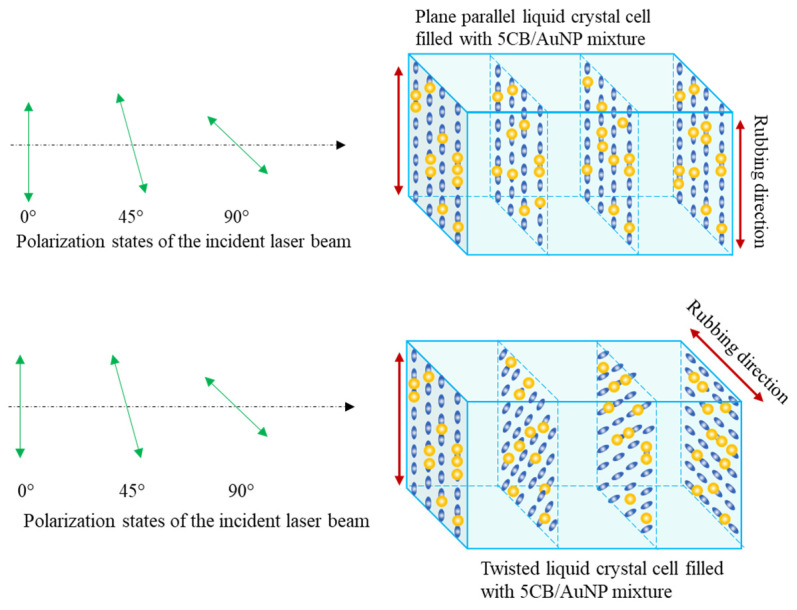
Schematic of the “vertical” arrangement of the liquid crystal cell with respect to the polarization of the incident light. Parallel (**top**) and twisted (**bottom**) nematic liquid crystal cells with AuNPs dispersed in them.

**Figure 3 micromachines-14-02206-f003:**
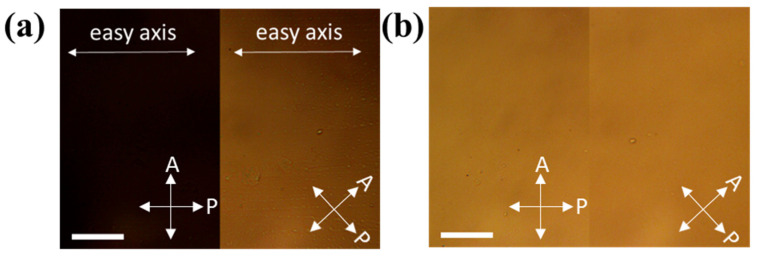
Cross-polarized and optical microscopy images of (**a**) oriented and (**b**) twisted cells. Angles of A—analyzer, and P—polarizer shown in lower right corners. (Scale bars are 50 μm).

**Figure 4 micromachines-14-02206-f004:**
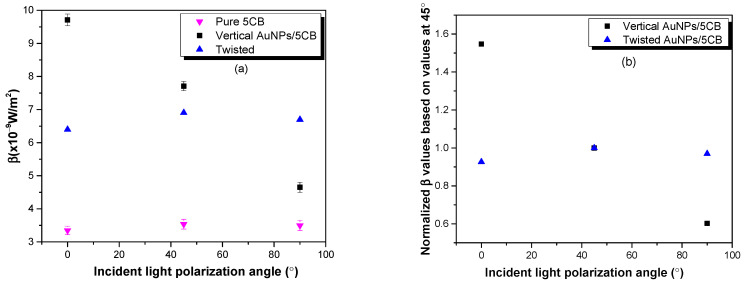
Plots show the absolute third-order nonlinear absorption coefficient (β) of 5CB and AuNP/5CB sample variation as the polarization of the incident light is varied. (**a**) 5CB (control, inverted triangle), AuNP/5CB mixture with plane parallel sample whose rubbing direction is vertical (square dots), and twisted sample (triangle dot); (**b**) shows the relative β values of vertical and twisted samples, where relative values are obtained from normalized absolute values based on β values at 45° in each group.

**Figure 5 micromachines-14-02206-f005:**
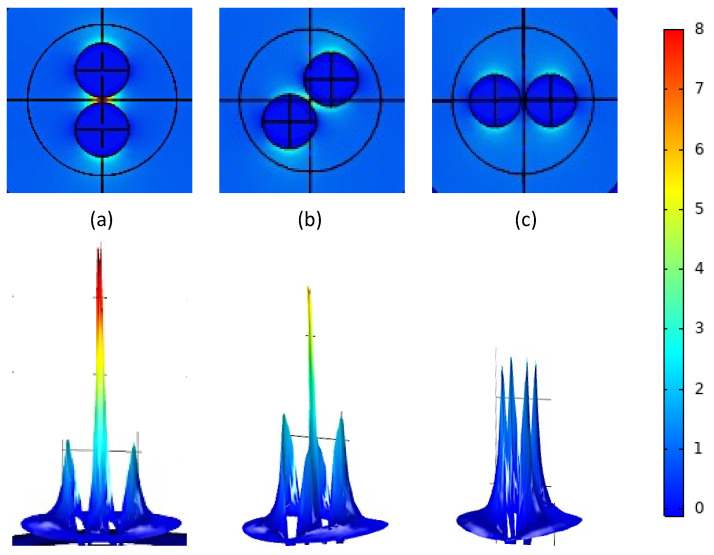
Field enhancement of a 1 V/m x-polarized plane wave incident along z onto the AuNP dimer with 2 nm separation. (**a**) Shows that orientation at 0° to polarization yields strong coupling and a large field enhancement; (**b**) shows dimer orientation at 45° with a small point of coupling in the middle; (**c**) shows dimer orientation at 90° with no coupling as the particles behave independently.

**Figure 6 micromachines-14-02206-f006:**
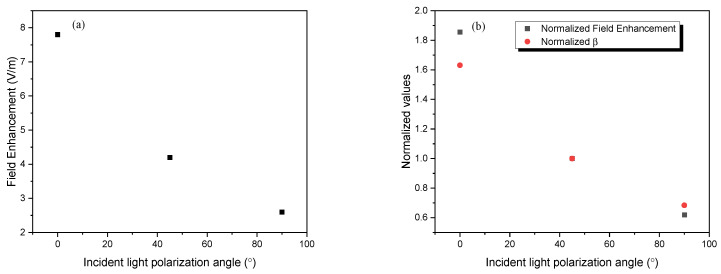
Plot showing (**a**) field enhancement vs. polarization angle, (**b**) field enhancement normalized to 45 vs. polarization angle, and then the normalized β vs. polarization angle.

**Figure 7 micromachines-14-02206-f007:**
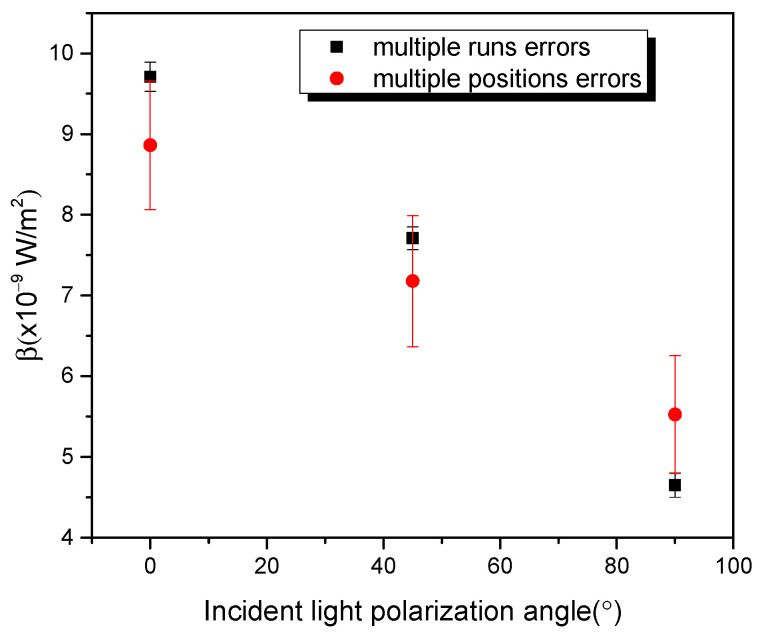
β values for one-position multiple measurements (runs) and multiple positions of oriented AuNPs/5CB samples when the laser polarization angle changed.

## Data Availability

The data used to support the findings of this study are included within the article.

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
