# Peer review of "Polarization Z-Scan Studies Revealing Plasmon Coupling Enhancement Due to Dimer Formation of Gold Nanoparticles in Nematic Liquid Crystals"

_micromachines, 2023, doi:10.3390/mi14122206_

Round 1

Reviewer 1 Report

Comments and Suggestions for Authors

The authors investigate the plasmon coupling of gold nanoparticle (AuNP) dimers dispersed in a nematic liquid crystal matrix using the polarization z-scan technique. The manuscript is novel and interesting. But some minor revisions about language should be made. Such as the mis-editting of "Vertical" to "Verticle" in Figure 4. 

Comments on the Quality of English Language

Some English language revisions should be made. 

Author Response

We thank the reviewer for finding our work “novel and interesting.” We corrected English and added references as per their suggestions.

Reviewer 2 Report

Comments and Suggestions for Authors

This manuscript reports on the control of the arrangement of gold nanoparticles (AuNPs) to experimentally examine their plasmonic coupling using the polarization z-scan technique. It sounds very challenging, and investigating these types of phenomena could provide interesting aspects to materials science and physical/computational chemistry.   However, a few important aspects should be included in the manuscript.

 It is highly recommended to include digital photos of AuNP solution as a function of preparation steps in the “Sample preparation” section.  Providing several photos of solution color according to each step would help readers understand the entire process (i.e., how AuNPs are treated and placed in analysis systems).    

 Based on the experimental conditions, there is no evidence demonstrating the actual alignment of AuNPs as the authors described (e.g., parallel liquid crystal cells by vertical rubbing direction and twisted liquid crystal cells by horizonal rubbing direction).  It would be much clearer if the authors could provide any evidence (e.g., SEM images).  This is because the plasmonic coupling could occur with different numbers of AuNPs.  This additional piece of results could strongly support the authors’ conclusions “The results demonstrate that in the case of parallel-oriented cells, when the incident light polarization aligns with the rubbing direction, the nonlinear absorption coefficient is approximately 2-3 times greater than that of 5CB alone. Additionally, it is observed that the value of the nonlinear absorption coefficient decreases linearly as the polarization angle changes from 0° to 90°. Conversely, for twisted nematic cells, the enhancement remains consistent across all polarization angles. These findings confirm that the enhancement of the nonlinear absorption coefficient is indeed attributed to plasmon coupling.”

Author Response

We thank the reviewer for the valuable comments. Response to their questions are provided below:

  1. It is highly recommended to include digital photos of AuNP solution as a function of preparation steps in the "Sample preparation" section. Providing several photos of solution color according to each step would help readers understand the entire process (i.e., how AuNPs are treated and placed in analysis systems).

Answer: Due to the low concentrations of AuNps that we are working with, there wasn't a significant visual difference in the colors of the mixtures with varying concentrations, as can be interpreted from UV-VIS that is provided in the supplementary section. 

  1. Based on the experimental conditions, there is no evidence demonstrating the actual alignment of AuNPs as the authors described (e.g., parallel liquid crystal cells by vertical rubbing direction and twisted liquid crystal cells by horizonal rubbing direction). It would be much clearer if the authors could provide any evidence (e.g., SEM images). This is because the plasmonic coupling could occur with different numbers of AuNPs. This additional piece of results could strongly support the authors' conclusions.

Answer: We thank Reviewer #2 for the suggestion. Measuring SEM with nanoparticles dispersed in complex materials like the nematic phase, which is a fluid, can be challenging. Nevertheless, aligning NPs within liquid crystals is not a new concept and has been demonstrated in many previous works, including our recent publication (Plasmon coupling assisted enhancement of third-order nonlinear optical absorption of gold nanoparticles dispersed in planar oriented nematic liquid crystals. Nanotechnology, 2023, 34, 365205). This alignment is directed by the elasticity of nematic LCs, which tends to orient the NPs along their directors to minimize the cost of elastic distortions and the system's free energy. In a uniform nematic, the nanoparticles (NPs) will form dimers along the direction of the director. If the nematic is twisted, the dimers will follow the liquid crystal orientation as well. Our study reveals a correlation between the Plasmon Coupling of the NPs and the orientation of the nematic director using the polarization Z-scan technique. We are demonstrating the potential of this technique to not only characterize the plasmonic properties of NPs but to also illustrate their alignment, which is new and promising in the field of liquid crystal/nanocomposites.